# QUASI-NEWTON POLICY GRADIENT ALGORITHMS

## ABSTRACT

Policy gradient algorithms have been widely applied to reinforcement learning (RL) problems in recent years. Regularization with various entropy functions is often used to encourage exploration and improve stability. In this paper, we propose a quasi-Newton method for the policy gradient algorithm with entropy regularization. In the case of Shannon entropy, the resulting algorithm reproduces the natural policy gradient (NPG) algorithm. For other entropy functions, this method results in brand new policy gradient algorithms. We provide a simple proof that all these algorithms enjoy the Newton-type quadratic convergence near the optimal policy. Using synthetic and industrial-scale examples, we demonstrate that the proposed quasi-Newton method typically converges in single-digit iterations, often orders of magnitude faster than other state-of-the-art algorithms.

## 1 INTRODUCTION

Consider an infinite-horizon MDP (Bellman, 1957; Sutton & Barto, 2018) $\mathcal{M} = (S, A, P, r, \gamma)$, where $S$ is a set of states of the system studied, $A$ is a set of actions made by the agent, $P$ is a transition probability tensor with $P_{st}^a$ being the probability of transitioning from state $s$ to state $t$ when taking action $a$, $r$ is a reward tensor with $r_s^a$ being the reward obtained when taking action $a$ at state $s$, and $0 < \gamma < 1$ is a discount factor. Throughout the paper, the state space $S$ and the action space $A$ are assumed to be finite. A policy $\pi$ is a randomized rule of action-selection where $\pi_s^a$ denotes the probability of choosing action $a$ at state $s$. For a given policy $\pi$, the value function $v_\pi$ defined as

$$(v_\pi)_s = \mathbb{E} \sum_{k=0}^{\infty} \left( \gamma^k r_{s_k}^{a_k} \mid s_0 = s \right) \tag{1}$$

satisfies the Bellman equation:

$$(I - \gamma P_\pi) v_\pi = r_\pi, \tag{2}$$

where $(P_\pi)_{st} = \sum_a \pi_s^a P_{st}^a$, $(r_\pi)_s = \sum_a \pi_s^a r_s^a$, and $I$ is the identity operator.

In order to promote exploration and enhance stability, one often regularizes the problem with a function $h_\pi$ such as the negative Shannon entropy $(h_\pi)_s = \sum_a \pi_s^a \log \pi_s^a$. With $h_\pi$, the original reward $r_\pi$ is replaced with the regularized reward $\tilde{r}_\pi = r_\pi - \tau h_\pi$ where $\tau$ is the regularization coefficient and (2) becomes

$$(I - \gamma P_\pi) v_\pi = \tilde{r}_\pi = r_\pi - \tau h_\pi, \tag{3}$$

where we overload the notation $v$ for the regularized value function. Other continuously differentiable entropy functions can be used as well, as we will show later. Since $\gamma < 1$ and $P_\pi$ is a transition probability matrix, $(I - \gamma P_\pi)$ is invertible, and

$$v_\pi = (I - \gamma P_\pi)^{-1} (r_\pi - \tau h_\pi). \tag{4}$$

In a policy optimization problem, we seek a policy $\pi$ that maximizes $e^\top v_\pi$ for some (in fact arbitrary) positive weight vector $e \in \mathbb{R}^{|S|}$. Using (4), the problem can be stated as

$$\max_\pi e^\top (I - \gamma P_\pi)^{-1} (r_\pi - \tau h_\pi). \tag{5}$$

This problem can be solved by, for example, the policy gradient (PG) method. However, the vanilla PG method converges quite slowly. In Agarwal et al. (2020), for example, the vanilla PG method is shown to have a $O(T^{-1})$ convergence rate, where $T$ denotes the number of iterations. For the

PG method with entropy regularization and some of its variants, the convergence rate can be improved to $O(e^{-cT})$, i.e., linear convergence (Mei et al., 2020), which can still be slow since $c$ is in general close to 0. It is also demonstrated in numerical examples that these algorithms with linear convergence rate can suffer from slow convergence. For example, in the example in Zhan et al. (2021), thousands of iterations are needed for the algorithm to converge, even though the model is relatively small and sparse. Therefore, there is a clear need for designing new methods with faster convergence and one idea is to take the geometry of the problem into consideration. The Newton method, for example, preconditions the gradient with the Hessian matrix and obtains second-order local convergence. Since the exact Hessian matrix is usually too computationally expensive to obtain, the quasi-Newton type methods, which use structurally simpler approximations of the Hessian instead, are more widely used in generic optimization problems, and are known to enjoy superlinear convergence (Rodomanov & Nesterov, 2021a;b).

## 1.1 CONTRIBUTIONS

In this paper, we investigate the quasi-Newton approach for solving (5). The main contributions of this paper are the following.

- First, we present a unified quasi-Newton type method for the policy optimization problem. The main observation is to decompose the Hessian as a sum of a diagonal part and a remainder, where at the optimal solution the remainder part vanishes so that the Hessian becomes diagonal. This inspires us to use only the diagonal part in the quasi-Newton method. As a result, the proposed method not only leverage the second order information but also enjoys low computational cost due to the diagonal structure of the preconditioner used. When the negative Shannon entropy is used, this method reproduces the natural policy gradient (NPG) algorithm. For other entropic regularizations, this method results in brand new policy gradient algorithms.

- Second, we analyze the convergence property of the proposed quasi-Newton algorithms and demonstrate local quadratic convergence both theoretically and numerically. By leveraging the quasi-Newton framework (Dennis & Moré, 1974), we provide a simple and straightforward proof for quadratic convergence near the optimal policy. In the numerical tests, we verify that the proposed method leads to fast quadratic convergence even under small regularization and large discount rate (close to 1). Even for industrial-size problems with hundreds of thousands states, the quasi-Newton method converges in single-digit iterations and within a few minutes on a regular laptop.

## 1.2 BACKGROUND AND RELATED WORK

A major workhorse behind the recent success in reinforcement learning (RL) is the large family of policy gradient (PG) methods (Williams, 1992; Sutton et al., 1999), for example, the natural policy gradient (NPG) method (Kakade, 2001), the actor-critic method (Konda & Tsitsiklis, 2000), the asynchronous advantage actor-critic (A3C) method (Mnih et al., 2016), the deterministic policy gradient (DPG) method (Silver et al., 2014), the trust region policy optimization (TRPO) (Schulman et al., 2015a), the generalized advantage estimation (GAE) (Schulman et al., 2015b), and proximal policy optimization (PPO) (Schulman et al., 2017), to mention but a few. The NPG method is known to be drastically faster than the original PG method, intuitively because in the NPG method the policy gradient is preconditioned by the Fisher information (an approximation of the Hessian of the KL-divergence) matrix and fits the problem geometry better. This idea is extended in TRPO and PPO where the problem geometry is taken into consideration via trust region constraints (in terms of KL-divergence) and a clipping function of the relative ratio of policies in the objective function, respectively. These implicit ways (in the sense that they do not adjust the gradient by an explicit preconditioner) of adjusting the policy gradient is in essence similar to the mirror descent (MD) method (Nemirovskij & Yudin, 1983) in generic optimization problems.

This similarity in addressing the inherent geometry of the problem is noticed by a line of recent work including Neu et al. (2017); Geist et al. (2019); Shani et al. (2020); Tomar et al. (2020); Lan (2021), and the analysis techniques in MD methods have been adapted to the PG setting. The connection was first built explicitly in Neu et al. (2017). The authors consider a linear program formulation where the objective function is the average reward and the domain is the set of stationary state-action

distributions, in which case the TRPO method can be viewed as an approximate mirror descent method and the A3C method as an MD method for the dual-averaging (Nesterov, 2009) objective. As a complement, Geist et al. (2019) considers an actor-critic type method where the policy is updated via either a regularized greedy step or an MD step and the value function is updated by a regularized Bellman operator, which also includes TRPO as a special case, and error propagation analysis is provided. In Shani et al. (2020), an adaptive scaling that naturally arises in the policy gradient is applied to the proximity term of the MD formulation, and sublinear convergence result is proved with a properly decreasing learning rate. In Tomar et al. (2020), the application to the non-tabular setting is enabled by parameterizing the policy and applying MD to the policy parameters, and the corresponding sublinear convergence result is presented.

Regularization, a strategy that considers the modified objective function with an additional penalty term on the policy, is another crucial component in the development of PG type methods. Intuitively, regularization is able to encourage exploration in the policy iteration process and thus avoid local minima. It is also suggested (Ahmed et al., 2019) that regularization makes the optimization landscape smoother and thus enables possibly faster convergence. Linear convergence result is then established for regularized PG and NPG methods (Agarwal et al., 2020; Mei et al., 2020; Cen et al., 2020). In these relatively earlier works (Agarwal et al., 2020; Mei et al., 2020; Cen et al., 2020), the regularization usually takes the form of (negative) entropy or relative entropy. In the more recent work Lan (2021) and Zhan et al. (2021) that follow the MD type methods, the regularization is extended to general convex functions with the resulting Bregman divergences different from the KL-divergence and linear convergence is guaranteed as well.

However, most of these algorithms are of either sublinear or linear convergence except the entropy regularized NPG with full step length (which is a special case of the quasi-Newton method we propose), and even the linear convergence rate $O(e^{-cT})$ can be slow since $c$ can be close to zero. This motivates us to invent the quasi-Newton policy gradient method to be introduced in Section 2.

## 2 QUASI-NEWTON METHOD

This section derives the quasi-Newton method for the entropic-regularized policy optimization problems. In this paper, we use a more general definition of the term "quasi-Newton method". Unlike classical quasi-Newton methods where the approximate Hessian matrix is constructed using first-order information, the proposed quasi-Newton method uses second-order information, but the approximate Hessian matrix remains simple and its inverse is easy to obtain. We start with the negative Shannon entropy $(h_\pi)_s = \sum_a \pi_s^a \log \pi_s^a$.

In what follows, assume that $\pi^*$ is the optimizer of the problem stated in (5). By introducing $Z_\pi := I - \gamma P_\pi$, the objective function can be written as

$$E(\pi) = e^\top (I - \gamma P_\pi)^{-1}(r_\pi - \tau h_\pi) = e^\top Z_\pi^{-1}(r_\pi - \tau h_\pi) = w_\pi^\top (r_\pi - \tau h_\pi), \qquad (6)$$

where $w_\pi := Z_\pi^{-\top} e$. For any $\epsilon$ with $\sum_a \epsilon_s^a = 0$, introduce $r_\epsilon \in \mathbb{R}^{|S|}$ and $Z_\epsilon \in \mathbb{R}^{|S| \times |S|}$ such that

$$(r_\epsilon)_s := \sum_a \epsilon_s^a r_s^a, \quad (Z_\epsilon)_{st} := \sum_a \epsilon_s^a (\delta_{st} - \gamma P_{st}^a),$$

where $\delta_{st} = 1$ if $s = t$ and $\delta_{st} = 0$ otherwise.

Let us first outline the main idea of the quasi-Newton method. The gradient of $E(\pi)$ in $\mathbb{R}^{|S||A|}$ is

$$\frac{\partial E}{\partial \pi_s^a} = (r_s^a - \tau(\log \pi_s^a + 1) - [(I - \gamma P^a)v_\pi]_s + c_s)(w_\pi)_s, \qquad (7)$$

where $c_s$ is the Lagrange multiplier associated with the constraint $\sum_a \pi_s^a = 1$. Our key observation is to decompose the Hessian matrix $D^2 E(\pi)$ in $\mathbb{R}^{|S||A| \times |S||A|}$ into two parts

$$D^2 E(\pi) = \tilde{H}(\pi) + H_r(\pi), \qquad (8)$$

where $\tilde{H}$ is a *diagonal* matrix given by $\tilde{H}_{(sa),(tb)} = -\tau \delta_{\{(sa),(tb)\}} \frac{(w_\pi)_s}{\pi_s^a}$ and $H_r$ is a remainder that *vanishes* at $\pi = \pi^*$, i.e., $H_r = O(\|\pi - \pi^*\|)$ (shown in Theorem 1). With this decomposition, we

can approximate the Hessian matrix $D^2 E(\pi)$ by $\tilde{H}$ and obtain the following *quasi-Newton flow*:

$$\frac{d\pi_s^a}{dt} = -(\tilde{H}^{-1}\nabla_\pi E)_{sa} = -(\tilde{H}_{(sa),(sa)})^{-1}\frac{\partial E}{\partial \pi_s^a}$$
$$= \pi_s^a(r_s^a - \tau(\log \pi_s^a + 1) - [(I - \gamma P^a)v_\pi]_s + c_s)/\tau,$$

By introducing the parameterization $\theta_s^a = \log \pi_s^a$ and discretizing in time with learning rate $\eta$, we arrive at

$$\theta_s^a \leftarrow \eta(r_s^a - \tau - [(I - \gamma P^a)v_\pi]_s + c_s)/\tau + (1 - \eta)\theta_s^a.$$

Writing this update back in terms of $\pi_s^a$ leads to the following update rule

$$\pi_s^a \propto (\pi_s^a)^{1-\eta}\exp(\eta(r_s^a + (\gamma P^a v_\pi)_s)/\tau).$$

This result is summarized in the following theorem with the proof given in Appendix A.

**Theorem 1.** *Let $h_\pi \in \mathbb{R}^{|S|}$ be the negative Shannon entropy $(h_\pi)_s = \sum_a \pi_s^a \log \pi_s^a$. (a) For any $\epsilon$ with $\sum_a \epsilon_s^a = 0$ and $|\epsilon_s^a| < \pi_s^a$, at $\pi = \pi^*$*

$$r_\epsilon - \tau Dh_\pi \epsilon - Z_\epsilon Z_\pi^{-1}(r_\pi - \tau h_\pi) = 0, \tag{9}$$

*where $Dh_\pi \in \mathbb{R}^{|S|\times|S||A|}$ is the gradient matrix of $h_\pi$ with respect to $\pi$. (b) There exists a diagonal approximation $\tilde{H}(\pi)$ of the Hessian matrix $D^2 E(\pi)$ such that*

$$\tilde{H}(\pi) - D^2 E(\pi) = O(\|\pi - \pi^*\|).$$

*(c) The quasi-Newton flow from $\tilde{H}(\pi)$ is*

$$\frac{d\pi_s^a}{dt} = \pi_s^a(r_s^a - \tau(\log \pi_s^a + 1) - [(I - \gamma P^a)v_\pi]_s + c_s)/\tau. \tag{10}$$

*With learning rate $\eta$, the gradient update is*

$$\pi_s^a \leftarrow \frac{(\pi_s^a)^{1-\eta}\exp(\eta(r_s^a + (\gamma P^a v_\pi)_s)/\tau)}{\sum_a(\pi_s^a)^{1-\eta}\exp(\eta(r_s^a + (\gamma P^a v_\pi)_s)/\tau)}. \tag{11}$$

**Remark 1.** *The policy update scheme (11) is the same as the entropy regularized natural policy gradient scheme in Cen et al. (2020).*

The conclusion in Theorem 1 can be extended to general entropies of the form

$$(h_\pi)_s = \sum_a \phi\left(\frac{\pi_s^a}{\mu_a}\right)\mu_a,$$

where $\phi$ is convex on $(0, +\infty)$ and $\phi(1) = 0$, and $\mu$ is a prior distribution over $A$. The term $(h_\pi)_s$ is also called the "$f$-divergence" between $\pi_s$ and $\mu$ (Rényi, 1961; Ali & Silvey, 1966). In what follows, we shall use the uniform prior unless otherwise specified. When $\phi(x) = x\log x$, $(h_\pi)_s = \sum_a\left(\frac{\pi_s^a}{\mu_a}\log\frac{\pi_s^a}{\mu_a}\right)\mu_a = \sum_a \pi_s^a \log \pi_s^a - \log\frac{1}{|A|}$ and by omitting the constant $\log\frac{1}{|A|}$ we recover the regularization used in Theorem 1. When $\phi(x) = \frac{4}{1-\alpha^2}(1 - x^{(1+\alpha)/2})$ $(\alpha < 1)$,

$$(h_\pi)_s = \frac{4}{|A|(1-\alpha^2)}\sum_a(1 - (|A|\pi_s^a)^{(1+\alpha)/2}) \tag{12}$$

is the $\alpha$-divergence. In particular, when $\alpha = 0$ we obtain the Hellinger divergence $(h_\pi)_s = -2\sum_a\sqrt{\frac{\pi_s^a}{|A|}}$ after omitting the constants and dividing by 2, and when $\alpha \to -1$ we obtain the reverse-KL divergence $(h_\pi)_s = \frac{1}{|A|}\sum_a \log\frac{1}{\pi_s^a}$ after omitting constants.

In the following theorem, we extend the quasi-Newton method in Theorem 1 to the entropy functions described above. The proof of this theorem can be found in Appendix B.

**Theorem 2.** *Assume that $\pi^*$ is the optimizer of (5) where $h_\pi$ denotes the $\alpha$-divergence (12). Then (a) the Hessian matrix $D^2 E(\pi)$ can be approximated by a diagonal matrix $\tilde{H}(\pi)$ near $\pi^*$ such that*

$\tilde{H}(\pi) - D^2 E(\pi) = O(\|\pi - \pi^*\|)$ *and (b) the quasi-Newton method from* $\tilde{H}(\pi)$ *(the policy gradient ascent preconditioned by* $-\tilde{H}(\pi)$*)) is*

$$\pi_s^a \leftarrow \left( (1-\eta)(\pi_s^a)^{\frac{\alpha-1}{2}} - \frac{1-\alpha}{2}\eta|A|^{\frac{1-\alpha}{2}}(r_s^a - [(I - \gamma P^a)v_\pi]_s)/\tau + c_s \right)^{\frac{2}{\alpha-1}}, \qquad (13)$$

*where* $0 < \eta \le 1$ *is the learning rate and with parameterization* $\theta_s^a = \phi'(|A|\pi_s^a)$, *the update scheme above can be expressed as:*

$$\theta_s^a \leftarrow \eta(r_s^a - [(I - \gamma P^a)v_\pi]_s + c_s)/\tau + (1-\eta)\theta_s^a. \qquad (14)$$

*where* $\phi(x) = \frac{4}{1-\alpha^2}(1 - x^{(1+\alpha)/2})$ ($\alpha < 1$) *and* $c_s$ *is the Lagrange multiplier introduced by the constraint* $\sum_a \pi_s^a = 1$.

**Remark 2.** *When* $\alpha = 0$ *(the Hellinger divergence), the corresponding update scheme is* $\pi_s^a \leftarrow \left( \frac{1-\eta}{\sqrt{\pi_s^a}} - \eta\sqrt{|A|}(r_s^a - [(I - \gamma P^a)v_\pi]_s)/\tau + c_s \right)^{-2}$, *which can be obtained by inserting* $\alpha = 0$ *in (13). When* $\alpha = -1$ *(the reverse-KL divergence), the corresponding update scheme is* $\pi_s^a \leftarrow \left( \frac{1-\eta}{\pi_s^a} - \eta|A|(r_s^a - [(I - \gamma P^a)v_\pi]_s)/\tau + c_s \right)^{-1}$, *which can be obtained by inserting* $\alpha = -1$.

**Remark 3.** *When a different prior* $\mu$ *is used, the corresponding algorithm can be obtained by inserting* $\theta_s^a = \phi'(\pi_s^a/\mu_a)$ *in (14).*

**Remark 4.** *All the policy update schemes can be extended to the case where* $\gamma = 1$ *and the MDP has a terminal state* $s_T$ *and the Markov chain is irreducible. In that case we can remove* $s_T$ *from* $S$ *and replace* $\gamma$ *with* 1 *in the update schemes. For example, the update scheme (11) becomes*

$$\pi_s^a \leftarrow \frac{(\pi_s^a)^{1-\eta}\exp(\eta(r_s^a + (\tilde{P}^a v_\pi)_s))}{\sum_a (\pi_s^a)^{1-\eta}\exp(\eta(r_s^a + (\tilde{P}^a v_\pi)_s))},$$

*where* $\tilde{P}$ *is the submatrix of* $P$ *with the row and column corresponding to* $s_T$ *removed.*

The remaining problem in the update schemes above is the determination of the multipliers $c_s$, since they cannot be solved explicitly as in the case of the negative Shannon entropy (Theorem 1). The determination of $c_s$ can be done in a similar way as in Ying (2020) based on the following lemma. The proof of this lemma can be found in Appendix C.

**Lemma 3.** *Assume that* $\sigma < 0$, *then for any* $x_1, x_2, \ldots, x_k$, *there is a unique solution to the equation:*

$$(x + x_1)^\sigma + \cdots + (x + x_k)^\sigma = 1, \qquad (15)$$

*such that* $x + x_i \ge 0$, $i = 1, 2, \ldots, k$. *Moreover, the solution is on the interval*

$$\left[ \max\left\{ -\min_{1 \le i \le k} x_i, k^{-\sigma} - \max_{1 \le j \le k} x_j \right\}, k^{-1/\sigma} - \min_{1 \le i \le k} x_i \right]. \qquad (16)$$

By Lemma 3 and the monotonicity of $(x + x_1)^\sigma + \cdots + (x + x_k)^\sigma - 1$, many of the established numerical methods (e.g. bisection) for nonlinear equations can be applied to determine the solution for (15). This routine can be used to find the multipliers $c_s$ as stated in Proposition 4 whose proof is given in Appendix D.

**Proposition 4.** *The multipliers* $c_s$ *in the update scheme (13) can be determined uniquely such that the updated policy* $\pi$ *satisfies* $\pi_s^a \ge 0$ *for any* $(s, a)$, *and* $\sum_a \pi_s^a = 1$ *for any* $s$.

The algorithm proposed in this section is summarized in Algorithm 1 below.

## 3 QUADRATIC CONVERGENCE

In this section, we study the quadratic convergence of the quasi-Newton method at learning rate $\eta = 1$. Our analysis is inspired by the results in Dennis & Moré (1974); Wang & Yan (2021). The following theorem states the second-order convergence when $\eta = 1$, with the proof given in Appendix E.

---

**Algorithm 1** Quasi-Newton method for the regularized MDP

---

**Require:** the MDP model $\mathcal{M} = (S, A, P, r, \gamma)$, initial policy $\pi_{\text{init}}$, convergence threshold $\epsilon_{\text{tol}}$, regularization coefficient $\tau$, learning rate $\eta$, the regularization type (KL, reverse-KL, Hellinger or $\alpha$-divergence).

1: Initialize the policy $\pi = \pi_{\text{init}}$.
2: Set $q = 1 + \epsilon_{\text{tol}}$ and $k = |A|$.
3: **while** $q > \epsilon_{\text{tol}}$ **do**
4:     Calculate the regularization term $h_\pi$ by $(h_\pi)_s = \frac{1}{|A|} \sum_a \phi(|A|\pi_s^a)$.
5:     Calculate $P_\pi$ and $r_\pi$ by $(P_\pi)_{st} = \sum_a \pi_s^a P_{st}^a, (r_\pi)_s = \sum_a \pi_s^a r_s^a$.
6:     Calculate $v_\pi$ by (4), i.e., $v_\pi = (I - \gamma P_\pi)^{-1}(r_\pi - \tau h_\pi)$.
7:     **if** the KL divergence is used **then**
8:        $(\pi_{\text{new}})_s^a \leftarrow \frac{(\pi_s^a)^{1-\eta} \exp(\eta(r_s^a + (\gamma P^a v_\pi)_s)/\tau)}{\sum_a (\pi_s^a)^{1-\eta} \exp(\eta(r_s^a + (\gamma P^a v_\pi)_s)/\tau)}$ for $a = 1, 2, \ldots, |A|,\ s = 1, 2, \ldots, |S|$.
9:     **end if**
10:    **if** the $\alpha$-divergence is used **then**
11:      **for** $s = 1, 2, \ldots, |S|$ **do**
12:        Set $\sigma = 2/(\alpha - 1)$.
13:        Calculate $x_a = (1-\eta)(\pi_s^a)^{\frac{\alpha-1}{2}} - \frac{1-\alpha}{2}\eta|A|^{\frac{1-\alpha}{2}}(r_s^a - [(I-\gamma P^a)v_\pi]_s)/\tau,\ a = 1, \ldots, |A|$.
14:        Solve for $c_s$ with the bisection method on the interval described in (16).
15:        Update $(\pi_{\text{new}})_s^a \leftarrow (c_s + x_a)^\sigma$ for $a = 1, 2, \ldots, |A|$.
16:      **end for**
17:    **end if**
18:    $q = \|\pi_{\text{new}} - \pi\|_F / \|\pi\|_F$.
19:    $\pi = \pi_{\text{new}}$
20: **end while**

---

**Theorem 5.** *Let $\eta = 1$, $f(\pi)_{sa} = -(r_s^a - ((I - \gamma P^a)v_\pi)_s)$ and*

$$\Phi_{KL}(\pi) = \sum_{sa} \pi_s^a \log \pi_s^a, \quad \Phi_{rKL}(\pi) = \frac{1}{|A|} \sum_{sa} \log \frac{1}{\pi_s^a},$$

$$\Phi_H(\pi) = -2 \sum_{sa} \sqrt{\frac{\pi_s^a}{|A|}}, \quad \Phi_\alpha(\pi) = \frac{4}{|A|(1-\alpha^2)} \sum_{sa} (1 - (|A|\pi_s^a)^{(1+\alpha)/2}), (\alpha < 1).$$

*Denote the $k$-th policy obtained in Algorithm 1 by $\pi^{(k)}$, then for $\eta = 1$ the update scheme in Algorithm 1 can be summarized as*

$$\nabla\Phi(\pi^{(k+1)}) - \nabla\Phi(\pi^{(k)}) = -\left( f(\pi^{(k)}) + \nabla\Phi(\pi^{(k)}) - B^\top c(\pi^{(k)}) \right), \tag{17}$$

*where we denote by $B$ the $|S|$-by-$(|S| \times |A|)$ matrix such that $B_{ij} = 1$ for $|A|(i-1)+1 \leq j \leq |A|i$ and $B_{ij} = 0$ otherwise. Define $(\theta^{(k)})_s^a = \phi'(|A|(\pi^{(k)})_s^a)$, where $\phi(x)$ is the convex function used in the regularization $(h_\pi)_s = \frac{1}{|A|} \sum_a \phi(|A|\pi_s^a)$. Assume that (1) $\nabla f$ is Lipschitz on a closed subset of $\{\pi : B^\top \pi = \mathbf{1}_{|S|}, \pi_s^a > 0\}$ and (2) $\theta^{(k)} \to \theta^*$. Then $\pi^{(k)}$ converges quadratically to $\pi^*$, i.e.,*

$$\|\pi^{(k+1)} - \pi^*\| \leq C\|\pi^{(k)} - \pi^*\|^2, \tag{18}$$

*for some constant $C$.*

**Connection with mirror descent** The quasi-Newton algorithm (17) for $\eta = 1$ has a deep connection with mirror descent. The vanilla mirror descent of $-E(\pi)$ with a learning rate $\beta$ and the Bregman divergence associated with $\Phi$ is given by

$$\pi^{(k+1)} = \arg\min_\pi \{-E(\pi^{(k)}) - \nabla E(\pi^{(k)})(\pi - \pi^{(k)}) + \frac{1}{\beta}(\Phi(\pi) - \Phi(\pi^{(k)}) - \nabla\Phi(\pi^{(k)})(\pi - \pi^{(k)}))\}$$

$$= \arg\min_\pi \{(\text{diag}(w_{\pi^{(k)}}) \otimes I_{|A|})(f(\pi^{(k)}) + \nabla\Phi(\pi^{(k)}))(\pi - \pi^{(k)}) + \frac{1}{\beta}(\Phi(\pi) - \nabla\Phi(\pi^{(k)})\pi)\}, \tag{19}$$

where $\mathrm{diag}(w_{\pi^{(k)}})$ is the diagonal matrix with the diagonal equal to $w_{\pi^{(k)}} := (I - \gamma P_{\pi^{(k)}}^\top)^{-1} e$, $\otimes$ denotes the Kronecker product, and $I_{|A|}$ denotes the identity matrix with size $|A|$ by $|A|$. In the last equality, the terms independent of $\pi$ are dropped and the multiplier term in $\nabla E$ is canceled out using $B\pi = B\pi^{(k)} = \mathbf{1}_{|S|}$. The first order stationary condition of this minimization problem reads

$$\nabla\Phi(\pi^{(k+1)}) - \nabla\Phi(\pi^{(k)}) = -\beta(\mathrm{diag}(w_{\pi^{(k)}}) \otimes I_{|A|})(f(\pi^{(k)}) + \nabla\Phi(\pi^{(k)}) - B^\top c(\pi^{(k)})). \quad (20)$$

This suggests that (17) can be reinterpreted as an *accelerated* mirror descent method with *adaptive* learning rates $\beta_s \equiv 1/(w_{\pi^{(k)}})_s$ that depend on the state $s$ and the current iterate $\pi^{(k)}$.

In Zhan et al. (2021), a variant of mirror descent is proposed based on an implicit update scheme

$$(\nabla\Phi(\pi^{(k+1)}))_{sa} - (\nabla\Phi(\pi^{(k)}))_{sa} = -\beta' \left( f(\pi^{(k)})_{sa} + \nabla\Phi(\pi^{(k+1)})_{sa} - (c(\pi^{(k)}))_s \right), \quad (21)$$

with a manually tuned learning rate $\beta'$. In the next section, we will compare this variant with our quasi-Newton method (17) in several examples and show that the quasi-Newton method converges orders of magnitudes faster than the ones in Zhan et al. (2021).

## 4 NUMERICAL EXPERIMENTS

### 4.1 EXPERIMENT I

We first test the quasi-Newton methods derived in Section 2 on the model in Zhan et al. (2021). For the sake of completeness, we include the model description here. The MDP considered has a state space $S$ of size $200$ and an action space $A$ of size $50$. For each state $t$ and action $a$, a subset $S_t^a$ of $S$ is uniformly randomly chosen such that $|S_t^a| = 20$, and $P_{tt'}^a = 1/20$ for any $t' \in S_t^a$. The reward is given by $r_s^a = U_s^a U_s$, where $U_s^a$ and $U_s$ are independently uniformly chosen on $[0, 1]$. The discount rate $\gamma$ is set as $0.99$ and the regularization coefficient $\tau = 0.001$.

In the numerical experiment, we implement Algorithm 1 with the KL divergence, the reverse KL divergence, the Hellinger divergence and the $\alpha$-divergence with $\alpha = -3$. We set the initial policy as the uniform policy, the convergence threshold as $\epsilon = 10^{-12}$ and the learning rate $\eta$ as 1. Figure 1(a) demonstrates that, for these four tests, the quasi-Newton algorithm converges in 7, 7, 7, and 6 iterations, respectively. In comparison, we apply the policy mirror descent (PMD) and the general policy mirror descent (GPMD) method in Zhan et al. (2021) to the same MDP with the same stopping criterion. As also shown in Figure 1(a), many more iterations are needed for GPMD and PMD to reach the same precision: GPMD converges in $14822$ iterations, and PMD does not reach the desired precision after $3 \times 10^5$ iterations.

In order to verify the quadratic convergence proved in Section 3, we draw the plots of $\log|\log\|\pi - \pi^*\|_F|$ in Figures 1(b), 1(c), 1(d) and 1(e), where $\pi^*$ is the final policy and the norm used is the Frobenius norm. A green reference line with slope $\log 2$ through the origin is plotted for comparison. If the error converges exactly at a quadratic rate, the plot of $\log|\log\|\pi - \pi^*\||$ shall be parallel to the reference line. The convergence curves approach the reference lines in the end (and are even steeper than the reference lines in the begining), demonstrating clearly a quadratic convergence for all regularizations used here.

### 4.2 EXPERIMENT II

Next, we apply the quasi-Newton methods derived in Section 2 to an MDP model constructed from the search logs of an online shopping store, with two different ranking strategies. Each issued query is represented as a state in the MDP. In response to a query, the search can be done by choosing one of the two ranking strategies (actions) to return a ranked list of products shown to the customer. Based on the shown products, the customer can refine or update the query, thus entering a new state. The reward at each state-action pair is a weighted sum of the clicks and purchases resulting from the action. Based on the data collected from two separate 5-week periods for both ranking strategies, we construct an MDP with 135k states, and a very sparse transition tensor $P$ with only $0.01\%$ nonzero entries. The discount rate $\gamma$ is set as $0.99$ and the regularization coefficient is $\tau = 0.001$.

When calculating $v_\pi$ by $v_\pi = (I - \gamma P_\pi)^{-1}(r_\pi - \tau h_\pi)$, we apply the iterative solver Bi-CGSTAB (Van der Vorst, 1992), a widely used numerical method with high efficiency and robustness for

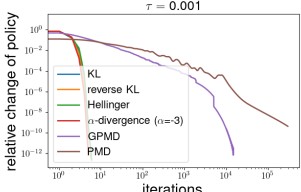 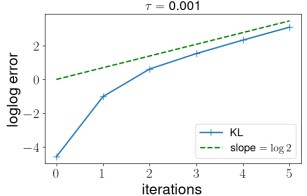 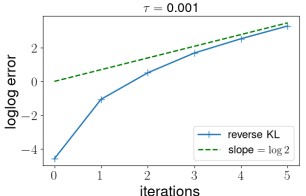

(a) Relative change of the policy using Algorithm 1 and methods from Zhan et al. (2021).

(b) The policy error in the process of training using KL-divergence.

(c) The policy error in the process of training using reverse KL-divergence.

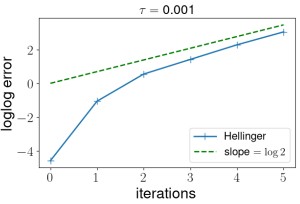 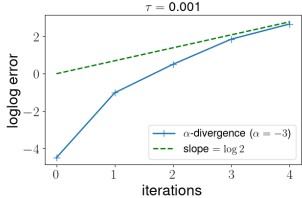

(d) The policy error in the process of training using Hellinger divergence.

(e) The policy error in the process of training using $\alpha$-divergence with $\alpha = -3$.

Figure 1: Figures for the synthetic medium scale MDP. (a): Relative change of the policy $\|\pi_{\text{new}} - \pi\|_F/\|\pi\|_F$ during training of Algorithm 1 compared with PMD and GPMD in Zhan et al. (2021), with the logarithmic scale used for both axes. Notice that Algorithm 1 converges in 6-7 iterations to $10^{-12}$ in all cases while PMD and GPMD take more than $10^4$ iterations. (b) - (e): Blue: The convergence of $\log|\log\|\pi - \pi^*\|_F|$ in the training process with the KL divergence, the reverse KL divergence, the Hellinger divergence and the $\alpha$-divergence with $\alpha = -3$, respectively. Green: A line through the origin with slope $\log 2$. Comparison of the convergence plots with the green reference lines shows a clear quadratic convergence for Algorithm 1.

solving large sparse nonsymmetric systems of linear equations (Saad, 2003; de Pillis, 1998), in order to leverage the sparsity of the transition tensor.

In the numerical experiment, we implement Algorithm 1 with the KL divergence, the reverse KL divergence, the Hellinger divergence and the $\alpha$-divergence with $\alpha = -3$. We set the initial policy as the uniform policy, the convergence threshold as $\epsilon = 10^{-12}$ and the learning rate $\eta$ as 1. All the tests end up with fast convergence as shown in Figure 2(a), where logarithmic scale is used for the vertical axis. More specifically, the quasi-Newton algorithm using the KL divergence, the reverse KL divergence, the Hellinger divergence and the $\alpha$-divergence with $\alpha = -3$ converge in $6, 6, 6, 5$ iterations, respectively. It is worth noticing that even though the size of the state space $S$ here is some magnitudes larger than the examples in Section 4.1, the number of quasi-Newton iterations used is about the same.

In Table 1, we report the number of BiCGSTAB steps used in the algorithm. In each quasi-Newton iteration, less than 20 BiCGSTAB steps are used in order to find $v_\pi$. For all the 4 regularizers used here, altogether only about 100 BiCGSTAB steps are needed in the whole training process, thanks to the fast convergence of the quasi-Newton method.

| Regularizer | KL | reverse-KL | Hellinger | $\alpha$-divergence ($\alpha = -3$) |
|---|---|---|---|---|
| Quasi-Newton Iterations | 6 | 6 | 6 | 5 |
| Total Bi-CGSTAB steps | 110 | 109 | 110 | 83 |
| Average Bi-CGSTAB steps | 18.3 | 18.2 | 18.3 | 16.6 |

Table 1: Number of quasi-Newton iterations and BiCGSTAB steps used in the training process.

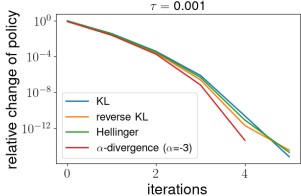 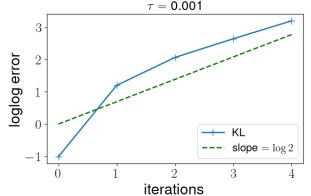 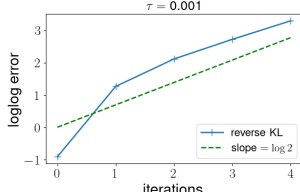

(a) Relative change of the policy $\|\pi_{\text{new}} - \pi\|_F / \|\pi\|_F$ in the training process.

(b) The policy error in the process of training using KL-divergence.

(c) The policy error in the process of training using reverse KL-divergence.

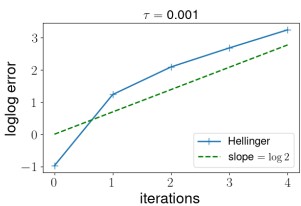 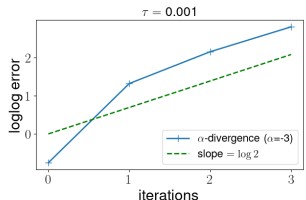

(d) The policy error in the process of training using Hellinger divergence.

(e) The policy error in the process of training using $\alpha$-divergence with $\alpha = -3$.

Figure 2: Figures for the synthetic medium scale MDP. (a): Relative change of the policy $\|\pi_{\text{new}} - \pi\|_F / \|\pi\|_F$ in the training process of Algorithm 1. Logarithmic scale is used for the vertical axis. (b) - (e): Blue: The convergence of $\log |\log \|\pi - \pi^*\|_F|$ in the training process with the KL divergence, the reverse KL divergence, the Hellinger divergence and the $\alpha$-divergence with $\alpha = -3$, respectively. Green: A line through the origin with slope $\log 2$.

As in the previous numerical example, in Figure 2(b), 2(c), 2(d) and 2(e) we verify the quadratic convergence by comparing the plot of $\log |\log \|\pi - \pi^*\||$ with a green reference line through the origin with slope $\log 2$. As the convergence curves are approximately parallel to the reference lines, this verifies that the proposed algorithm converges quadratically with all the regularizations in this example as well.

## 5 DISCUSSION

In this paper, we present a fast quasi-Newton method for the policy gradient algorithm that is capable of solving large MDP problems with large ($\approx 1$) discount rate and small ($\approx 0$) regularizations. The proposed method includes the well-known natural policy gradient algorithm as a special case, and naturally extends to other regularizers such as the reverse KL divergence, the Hellinger divergence and the $\alpha$-divergence.

Theoretically, we show local quadratic convergence of the proposed quasi-Newton method with a relatively simple proof. This quadratic convergence is confirmed numerically on both medium and large sparse models. In contrast with mirror descent type methods (e.g. Zhan et al. (2021)) that take up to tens of thousands iterations even with manually tuned learning rate, the proposed quasi-Newton algorithms typically converge in less than 10 iterations, despite the large discount rate (close to 1) and small regularization coefficient ($\tau \approx 0$).

For future work, we plan to adapt the technique used here to other gradient based algorithms for solving the MDP problems. Other types of $f$-divergence can also be included. Another direction is to consider continuous MDP problems by leveraging function approximation, effective spatial discretization, or model reduction.

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

# A    PROOF OF THEOREM 1

**Theorem 1.** *Let $h_\pi \in \mathbb{R}^{|S|}$ be the negative Shannon entropy $(h_\pi)_s = \sum_a \pi_s^a \log \pi_s^a$. (a) For any $\epsilon$ with $\sum_a \epsilon_s^a = 0$ and $|\epsilon_s^a| < \pi_s^a$, at $\pi = \pi^*$*

$$r_\epsilon - \tau Dh_\pi \epsilon - Z_\epsilon Z_\pi^{-1}(r_\pi - \tau h_\pi) = 0, \tag{9}$$

*where $Dh_\pi \in \mathbb{R}^{|S| \times |S||A|}$ is the gradient matrix of $h_\pi$ with respect to $\pi$. (b) There exists a diagonal approximation $\tilde{H}(\pi)$ of the Hessian matrix $D^2 E(\pi)$ such that*

$$\tilde{H}(\pi) - D^2 E(\pi) = O(\|\pi - \pi^*\|).$$

*(c) The quasi-Newton flow from $\tilde{H}(\pi)$ is*

$$\frac{d\pi_s^a}{dt} = \pi_s^a(r_s^a - \tau(\log \pi_s^a + 1) - [(I - \gamma P^a)v_\pi]_s + c_s)/\tau. \tag{10}$$

*With learning rate $\eta$, the gradient update is*

$$\pi_s^a \leftarrow \frac{(\pi_s^a)^{1-\eta} \exp(\eta(r_s^a + (\gamma P^a v_\pi)_s)/\tau)}{\sum_a (\pi_s^a)^{1-\eta} \exp(\eta(r_s^a + (\gamma P^a v_\pi)_s)/\tau)}. \tag{11}$$

*Proof.* Since $\pi$ is a policy, $\sum_a \pi_s^a = 1$ for any $s$. Thus

$$(Z_\pi)_{st} = \delta_{st} - \gamma \sum_a \pi_s^a P_{st}^a = \sum_a \pi_s^a(\delta_{st} - \gamma P_{st}^a). \tag{22}$$

Now consider a policy $\pi + \epsilon$ close to $\pi$, i.e., $\sum_a \epsilon_s^a = 0$ and $|\epsilon_s^a| \ll \pi_s^a$, then by (22),

$$Z_{\pi+\epsilon} = Z_\pi + Z_\epsilon, \quad r_{\pi+\epsilon} = r_\pi + r_\epsilon. \tag{23}$$

With (23), we have

$$
\begin{aligned}
&E(\pi + \epsilon) \\
&= e^\top Z_{\pi+\epsilon}^{-1}(r_{\pi+\epsilon} - \tau h_{\pi+\epsilon}) \\
&= e^\top (Z_\pi + Z_\epsilon)^{-1}(r_\pi + r_\epsilon - \tau h_{\pi+\epsilon}) \\
&= e^\top (Z_\pi^{-1} - Z_\pi^{-1} Z_\epsilon Z_\pi^{-1} + Z_\pi^{-1} Z_\epsilon Z_\pi^{-1} Z_\epsilon Z_\pi^{-1})(r_\pi + r_\epsilon - \tau h_\pi - \tau Dh_\pi \epsilon - \frac{1}{2}\epsilon^\top \tau D^2 h_\pi \epsilon) + o(\epsilon^2) \\
&\approx e^\top Z_\pi^{-1}(r_\pi - \tau h_\pi) + e^\top \left[ -Z_\pi^{-1} Z_\epsilon Z_\pi^{-1}(r_\pi - \tau h_\pi) + Z_\pi^{-1}(r_\epsilon - \tau Dh_\pi \epsilon) \right] \\
&+ e^\top \left[ Z_\pi^{-1}(-\frac{1}{2}\epsilon^\top \tau D^2 h_\pi \epsilon) + (-Z_\pi^{-1} Z_\epsilon Z_\pi^{-1})(r_\epsilon - \tau Dh_\pi \epsilon) + (Z_\pi^{-1} Z_\epsilon Z_\pi^{-1} Z_\epsilon Z_\pi^{-1})(r_\pi - \tau h_\pi) \right],
\end{aligned} \tag{24}
$$

where we omit the $O(\|\epsilon\|^3)$ term and keep the first two orders in the last step, and $Dh_\pi$ is a second-order tensor that maps from $S \times A$ to $S$, and $D^2 h_\pi$ is a third-order tensor that maps from $(S \times A)^{\otimes 2}$ to $S$. With this expansion, we can see that

$$\frac{\partial E}{\partial \pi_s^a} = (r_s^a - \tau(\log \pi_s^a + 1) - [(I - \gamma P^a)v_\pi]_s + c_s)(w_\pi)_s, \tag{25}$$

where $w_\pi = Z_\pi^{-\top} e$ and $c_s(w_\pi)_s$ is the Lagrange multiplier. Then at $\pi = \pi^*$,

$$\frac{\partial E}{\partial \pi_s^a} = (r_s^a - \tau(\log \pi_s^a + 1) - [(I - \gamma P^a)v_\pi]_s + c_s)(w_\pi)_s = 0.$$

Since $w_\pi = (I - \gamma P_\pi^\top)^{-1} e = e + \sum_{i=1}^\infty \gamma^i (P_\pi^\top)^i e$ and all elements of $e$ are positive, we also know that all elements of $w_\pi$ are positive, thus at $\pi = \pi^*$,

$$r_s^a - \tau(\log \pi_s^a + 1) - [(I - \gamma P^a)v_\pi]_s + c_s = 0.$$

Multiplying the left hand side with $\epsilon_s^a$ and taking the sum over $a$ we obtain:

$$(r_\epsilon - \tau Dh_\pi \epsilon - Z_\epsilon v_\pi)_s + c_s \sum_a \epsilon_s^a = 0, \quad \forall s, \quad \forall \epsilon,$$

and since $\sum_a \epsilon_s^a = 0$ for any $s$ and $v_\pi = Z_\pi^{-1}(r_\pi - \tau h_\pi)$, we have

$$r_\epsilon - \tau D h_\pi \epsilon - Z_\epsilon Z_\pi^{-1}(r_\pi - \tau h_\pi) = 0, \quad \forall \epsilon,$$

at $\pi = \pi^*$, which proves (9).

We can now simplify the second-order term in (24) by approximating $r_\epsilon - \tau D h_\pi \epsilon - Z_\epsilon Z_\pi^{-1}(r_\pi - \tau h_\pi)$ with 0 for $\pi$ near $\pi^*$,

$$
\begin{aligned}
&e^\top \left[ Z_\pi^{-1}(-\frac{1}{2}\epsilon^\top \tau D^2 h_\pi \epsilon) + (-Z_\pi^{-1} Z_\epsilon Z_\pi^{-1})(r_\epsilon - \tau D h_\pi \epsilon) + (Z_\pi^{-1} Z_\epsilon Z_\pi^{-1} Z_\epsilon Z_\pi^{-1})(r_\pi - \tau h_\pi) \right] \\
&= e^\top \left[ Z_\pi^{-1}(-\frac{1}{2}\epsilon^\top \tau D^2 h_\pi \epsilon) - Z_\pi^{-1} Z_\epsilon Z_\pi^{-1} \left( r_\epsilon - \tau D h_\pi \epsilon - Z_\epsilon Z_\pi^{-1}(r_\pi - \tau h_\pi) \right) \right] \\
&\approx e^\top Z_\pi^{-1}(-\frac{1}{2}\epsilon^\top \tau D^2 h_\pi \epsilon) = \frac{1}{2}\epsilon^\top \tilde{H}(\pi)\epsilon.
\end{aligned}
$$

$$(26)$$

By (9) and the twice continuous differentiability of $h$, it is clear that the approximate Hessian $\tilde{H}$ converge to the true Hessian as $\pi$ converges to $\pi^*$, and $\tilde{H}(\pi) - D^2 E(\pi) = O(\|\pi - \pi^*\|)$. The second-order derivatives are approximately given by

$$\frac{\partial^2 E}{\partial \pi_s^a \partial \pi_t^b} \approx \tilde{H}_{(sa),(tb)} = -\tau \delta_{\{(sa),(tb)\}} \frac{(w_\pi)_s}{\pi_s^a}, \tag{27}$$

from which we have shown that $\tilde{H}$ is diagonal.

Using this approximate second-order derivative as preconditioner, $w_\pi$ is canceled out in the policy gradient algorithm, which becomes

$$\frac{d\pi_s^a}{dt} = \pi_s^a (r_s^a - \tau(\log \pi_s^a + 1) - [(I - \gamma P^a)v_\pi]_s + c_s)/\tau.$$

Adopting the parameterization $\pi_s^a = \exp(\theta_s^a)$, we have

$$\frac{d\theta_s^a}{dt} = (r_s^a - \tau(\theta_s^a + 1) - [(I - \gamma P^a)v_\pi]_s + c_s)/\tau. \tag{28}$$

With a learning rate $\eta$, this becomes

$$\theta_s^a \leftarrow \eta(r_s^a - \tau - [(I - \gamma P^a)v_\pi]_s + c_s)/\tau + (1 - \eta)\theta_s^a, \tag{29}$$

which corresponds to

$$\pi_s^a \leftarrow (\pi_s^a)^{1-\eta} \exp(\eta(r_s^a - \tau - [(I - \gamma P^a)v_\pi]_s + c_s)/\tau), \tag{30}$$

and $c_s$ is determined by the condition that $\sum_a \pi_s^a = 1$ for any $s$. Equivalently, we have

$$\pi_s^a \leftarrow \frac{(\pi_s^a)^{1-\eta} \exp(\eta(r_s^a - \tau - [(I - \gamma P^a)v_\pi]_s)/\tau)}{\sum_a (\pi_s^a)^{1-\eta} \exp(\eta(r_s^a - \tau - [(I - \gamma P^a)v_\pi]_s)/\tau)} = \frac{(\pi_s^a)^{1-\eta} \exp(\eta(r_s^a + (\gamma P^a v_\pi)_s)/\tau)}{\sum_a (\pi_s^a)^{1-\eta} \exp(\eta(r_s^a + (\gamma P^a v_\pi)_s)/\tau)},$$

where we cancel out the factors independent of $a$. This is exactly (11), which finishes the proof. $\quad\square$

## B  PROOF OF THEOREM 2

**Theorem 2.** *Assume that $\pi^*$ is the optimizer of (5) where $h_\pi$ denotes the $\alpha$-divergence (12). Then (a) the Hessian matrix $D^2 E(\pi)$ can be approximated by a diagonal matrix $\tilde{H}(\pi)$ near $\pi^*$ such that $\tilde{H}(\pi) - D^2 E(\pi) = O(\|\pi - \pi^*\|)$ and (b) the quasi-Newton method from $\tilde{H}(\pi)$ (the policy gradient ascent preconditioned by $-\tilde{H}(\pi)$) is*

$$\pi_s^a \leftarrow \left( (1-\eta)(\pi_s^a)^{\frac{\alpha-1}{2}} - \frac{1-\alpha}{2}\eta|A|^{\frac{1-\alpha}{2}} (r_s^a - [(I - \gamma P^a)v_\pi]_s)/\tau + c_s \right)^{\frac{2}{\alpha-1}}, \tag{13}$$

*where $0 < \eta \leq 1$ is the learning rate and with parameterization $\theta_s^a = \phi'(|A|\pi_s^a)$, the update scheme above can be expressed as:*

$$\theta_s^a \leftarrow \eta(r_s^a - [(I - \gamma P^a)v_\pi]_s + c_s)/\tau + (1 - \eta)\theta_s^a. \tag{14}$$

*where $\phi(x) = \frac{4}{1-\alpha^2}(1 - x^{(1+\alpha)/2})$ $(\alpha < 1)$ and $c_s$ is the Lagrange multiplier introduced by the constraint $\sum_a \pi_s^a = 1$.*

*Proof.* Since the only difference between the functional $E(\pi)$ defined here and the $E(\pi)$ in Theorem 1 lies in the regularizer $h$, we still have

$$w_\pi^\top \left( r_\epsilon - \tau D h_\pi \epsilon - Z_\epsilon Z_\pi^{-1}(r_\pi - \tau h_\pi) \right) = 0, \quad \forall \epsilon,$$

at $\pi = \pi^*$, where $w_\pi = (I - \gamma P_\pi^\top)^{-1} e$. Moreover, we also have

$$\frac{\partial E}{\partial \pi_s^a} = (r_s^a - \tau \phi'(|A|\pi_s^a) - [(I - \gamma P^a)v_\pi]_s + c_s)(w_\pi)_s, \tag{31}$$

where $c_s(w_\pi)_s$ is the Lagrange multiplier. Since $w_\pi = e + \sum_{i=1}^\infty \gamma^i (P_\pi^\top)^i e$ and all elements of $e$ are positive, all elements of $w$ are positive as well. Thus at $\pi = \pi^*$,

$$(r_s^a - \tau \phi'(|A|\pi_s^a) - [(I - \gamma P^a)v_\pi]_s + c_s) = 0, \tag{32}$$

which leads to

$$r_\epsilon - \tau D h_\pi \epsilon - Z_\epsilon Z_\pi^{-1}(r_\pi - \tau h_\pi) = 0, \quad \forall \epsilon,$$

and the approximation:

$$E(\pi + \epsilon) - E(\pi) = w_\pi^\top (-\frac{1}{2}\epsilon^\top \tau D^2 h_\pi \epsilon) + O(\|\epsilon\|^3) = \frac{1}{2}\epsilon^\top \tilde{H}(\pi)\epsilon + O(\|\epsilon\|^3). \tag{33}$$

Hence we have proved that $D^2 E(\pi) - \tilde{H}(\pi) = O(\|\epsilon\|)$. With this approximation, we have

$$\frac{\partial^2 E}{\partial \pi_s^a \partial \pi_t^b} \approx \tilde{H}_{(sa),(tb)} = -\tau \delta_{\{(sa),(tb)\}}(w_\pi)_s |A|\phi''(|A|\pi_s^a), \tag{34}$$

Which shows that $\tilde{H}$ is diagonal. Then the policy gradient scheme becomes

$$\frac{\mathrm{d}\pi_s^a}{\mathrm{d}t} = (|A|\phi''(|A|\pi_s^a))^{-1}(r_s^a - \tau \phi'(|A|\pi_s^a) - [(I - \gamma P^a)v_\pi]_s + c_s)/\tau, \tag{35}$$

or equivalently,

$$\frac{\mathrm{d}(\phi'(|A|\pi_s^a))}{\mathrm{d}t} = (r_s^a - \tau \phi'(|A|\pi_s^a) - [(I - \gamma P^a)v_\pi]_s + c_s)/\tau. \tag{36}$$

Let $\theta_s^a = \phi'(|A|\pi_s^a)$, then

$$\frac{\mathrm{d}\theta_s^a}{\mathrm{d}t} = (r_s^a - \tau \theta_s^a - [(I - \gamma P^a)v_\pi]_s + c_s)/\tau,$$

With a learning rate $\eta$, this becomes

$$\theta_s^a \leftarrow \eta(r_s^a - [(I - \gamma P^a)v_\pi]_s + c_s)/\tau + (1 - \eta)\theta_s^a,$$

which is exactly (14). We derive the quasi-Newton schemes by direct calculations below.

1. When $(h_\pi)_s = \frac{1}{|A|}\sum_a \log \frac{1}{\pi_s^a}$, we have $\theta_s^a = -\frac{1}{|A|\pi_s^a}$, then (14) corresponds to

$$\pi_s^a \leftarrow \left( \frac{1-\eta}{\pi_s^a} - \eta|A|(r_s^a - [(I - \gamma P^a)v_\pi]_s + c_s)/\tau \right)^{-1},$$

   and $c_s$ is determined by the condition that $\sum_a \pi_s^a = 1$ for any $s$.

2. When $(h_\pi)_s = -2\sum_a \sqrt{\frac{\pi_s^a}{|A|}}$, we have $\theta_s^a = -\frac{1}{\sqrt{|A|\pi_s^a}}$, then (14) corresponds to

$$\pi_s^a \leftarrow \left( \frac{1-\eta}{\sqrt{\pi_s^a}} - \eta\sqrt{|A|}(r_s^a - [(I - \gamma P^a)v_\pi]_s)/\tau + c_s \right)^{-2},$$

   and $c_s$ is determined by the condition that $\sum_a \pi_s^a = 1$ for any $s$.

3. When $(h_\pi)_s = \frac{4}{|A|(1-\alpha^2)}\sum_a(1 - (|A|\pi_s^a)^{(1+\alpha)/2})$, we have $\theta_s^a = -\frac{2}{1-\alpha}(|A|\pi_s^a)^{\frac{\alpha-1}{2}}$, then (14) corresponds to

$$\pi_s^a \leftarrow \left( (1-\eta)(\pi_s^a)^{\frac{\alpha-1}{2}} - \frac{1-\alpha}{2}\eta|A|^{\frac{1-\alpha}{2}}(r_s^a - [(I - \gamma P^a)v_\pi]_s)/\tau + c_s \right)^{\frac{2}{\alpha-1}},$$

   and $c_s$ is determined by the condition that $\sum_a \pi_s^a = 1$ for any $s$.

$\square$

## C    PROOF OF LEMMA 3

**Lemma 3.** *Assume that $\sigma < 0$, then for any $x_1, x_2, \ldots, x_k$, there is a unique solution to the equation:*

$$(x + x_1)^\sigma + \cdots + (x + x_k)^\sigma = 1, \tag{15}$$

*such that $x + x_i \geq 0$, $i = 1, 2, \ldots, k$. Moreover, the solution is on the interval*

$$\left[ \max \left\{ - \min_{1 \leq i \leq k} x_i, k^{-\sigma} - \max_{1 \leq j \leq k} x_j \right\}, k^{-1/\sigma} - \min_{1 \leq i \leq k} x_i \right]. \tag{16}$$

*Proof.* Let

$$f(x) = (x + x_1)^\sigma + \cdots + (x + x_k)^\sigma,$$

then $f(x)$ is positive and decreasing on $(- \min_{1 \leq i \leq k} x_i, \infty)$ since each summand is positive and decreasing. When $x \to - \min_{1 \leq i \leq k} x_i$ from the right, $f(x) \to +\infty$ since at least one of the summand goes to $+\infty$. If $k^{-1/\sigma} - \max_{1 \leq i \leq k} x_i \geq - \min_{1 \leq i \leq k} x_i$, then when $x = k^{-1/\sigma} - \max_{1 \leq i \leq k} x_i$,

$$f(x) = \sum_{i=1}^{k} (k^{-1/\sigma} - \max_{1 \leq j \leq k} x_j + x_i)^\sigma \geq \sum_{i=1}^{k} (k^{-1/\sigma})^\sigma = k \times \frac{1}{k} = 1.$$

Moreover, when $x = k^{-1/\sigma} - \min_{1 \leq i \leq k} x_i$,

$$f(x) = \sum_{i=1}^{k} (k^{-1/\sigma} - \min_{1 \leq j \leq k} x_j + x_i)^\sigma \leq \sum_{i=1}^{k} (k^{-1/\sigma})^\sigma = k \times \frac{1}{k} = 1.$$

By the continuity of $f$, there exists a solution $x$ to (15) on

$$\left[ \max\{ - \min_{1 \leq i \leq k} x_i, k^{-1/\sigma} - \max_{1 \leq j \leq k} x_j \}, k^{-1/\sigma} - \min_{1 \leq i \leq k} x_i \right],$$

and the solution is unique by the monotonicity of $f$ on $(- \min_{1 \leq i \leq k} x_i, \infty)$. $\square$

## D    PROOF OF PROPOSITION 4

**Proposition 4.** *The multipliers $c_s$ in the update scheme (13) can be determined uniquely such that the updated policy $\pi$ satisfies $\pi_s^a \geq 0$ for any $(s, a)$, and $\sum_a \pi_s^a = 1$ for any s.*

*Proof.* For all the three update schemes, we apply the routine stated in Lemma 3 with $k = |A|$ for $|S|$ times in each iteration. For the reverse-KL divergence, $c_s$ is the solution to (15) for $\sigma = -1$ and

$$x_a = \frac{1 - \eta}{\pi_s^a} - \eta |A|(r_s^a - [(I - \gamma P^a) v_\pi]_s)/\tau, \quad a = 1, 2, \ldots, |A|.$$

For the Hellinger divergence, $c_s$ is the solution to (15) for $\sigma = -2$ and

$$x_a = \frac{1 - \eta}{\sqrt{\pi_s^a}} - \eta \sqrt{|A|}(r_s^a - [(I - \gamma P^a) v_\pi]_s)/\tau, \quad a = 1, 2, \ldots, |A|.$$

For the update scheme (13), $c_s$ is the solution to (15) for $\sigma = 2/(\alpha - 1)$ and

$$x_a = (1 - \eta)(\pi_s^a)^{\frac{\alpha - 1}{2}} - \frac{1 - \alpha}{2} \eta |A|^{\frac{1 - \alpha}{2}}(r_s^a - [(I - \gamma P^a) v_\pi]_s)/\tau, \quad a = 1, 2, \ldots, |A|.$$

By Lemma 3, the solution to the multipliers such that the updated $\pi_s^a \geq 0$ is unique, and the constraint $\sum_a \pi_s^a = 1$ is satisfied since $c_s$ is the solution to (15) (with the $x_i$-s defined above) and the right hand side of the (15) is 1. $\square$

# E PROOF OF THEOREM 5

**Theorem 5.** *Let $\eta = 1$, $f(\pi)_{sa} = -(r_s^a - ((I - \gamma P^a)v_\pi)_s)$ and*

$$\Phi_{KL}(\pi) = \sum_{sa} \pi_s^a \log \pi_s^a, \quad \Phi_{rKL}(\pi) = \frac{1}{|A|} \sum_{sa} \log \frac{1}{\pi_s^a},$$

$$\Phi_H(\pi) = -2 \sum_{sa} \sqrt{\frac{\pi_s^a}{|A|}}, \quad \Phi_\alpha(\pi) = \frac{4}{|A|(1-\alpha^2)} \sum_{sa} (1 - (|A|\pi_s^a)^{(1+\alpha)/2}), (\alpha < 1).$$

*Denote the $k$-th policy obtained in Algorithm 1 by $\pi^{(k)}$, then for $\eta = 1$ the update scheme in Algorithm 1 can be summarized as*

$$\nabla\Phi(\pi^{(k+1)}) - \nabla\Phi(\pi^{(k)}) = -\left( f(\pi^{(k)}) + \nabla\Phi(\pi^{(k)}) - B^\top c(\pi^{(k)}) \right), \tag{17}$$

*where we denote by $B$ the $|S|$-by-$(|S| \times |A|)$ matrix such that $B_{ij} = 1$ for $|A|(i-1)+1 \leq j \leq |A|i$ and $B_{ij} = 0$ otherwise. Define $(\theta^{(k)})_s^a = \phi'(|A|(\pi^{(k)})_s^a)$, where $\phi(x)$ is the convex function used in the regularization $(h_\pi)_s = \frac{1}{|A|} \sum_a \phi(|A|\pi_s^a)$. Assume that (1) $\nabla f$ is Lipschitz on a closed subset of $\{\pi : B^\top \pi = \mathbf{1}_{|S|}, \pi_s^a > 0\}$ and (2) $\theta^{(k)} \to \theta^*$. Then $\pi^{(k)}$ converges quadratically to $\pi^*$, i.e.,*

$$\|\pi^{(k+1)} - \pi^*\| \leq C\|\pi^{(k)} - \pi^*\|^2, \tag{18}$$

*for some constant $C$.*

*Proof.* Since $B_{ij} = 1$ for $|A|(i-1)+1 \leq j \leq |A|i$ and $B_{ij} = 0$ otherwise, the constraint on $\pi$ can be expressed by $B\pi = \mathbf{1}_{|S|}$. The gradient flow (28) can thus be expressed by

$$\frac{\mathrm{d}(\nabla\Phi(\pi))}{\mathrm{d}t} = -(f(\pi) + \nabla\Phi(\pi) - B^\top c(\pi)), \tag{37}$$

where $c(\pi)$ is the Lagrange multiplier. The corresponding discrete scheme used is hence

$$\nabla\Phi(\pi^{(k+1)}) - \nabla\Phi(\pi^{(k)}) = -\left( f(\pi^{(k)}) + \nabla\Phi(\pi^{(k)}) - B^\top c(\pi^{(k)}) \right). \tag{38}$$

With this update scheme, we have

$$f(\pi^{(k+1)}) - f(\pi^{(k)}) - \nabla f(\pi^*) \left( \pi^{(k+1)} - \pi^{(k)} \right)$$
$$= f(\pi^{(k+1)}) + \nabla\Phi(\pi^{(k+1)}) - B^\top c(\pi^{(k)}) - \nabla f(\pi^*) \left( \pi^{(k+1)} - \pi^{(k)} \right) \tag{39}$$

Since $\theta^{(k)} = \nabla\Phi(\pi^{(k)})$ converges to $\theta^*$ and the map from $\theta$ to $\pi$ is continuous, we have $\pi$ converges to some $\pi^*$ as well. Since $\pi^{(k)} \in \{\pi : B^\top \pi = \mathbf{1}_{|S|}, \pi_s^a \geq 0\}$ and $\{\pi : B^\top \pi = \mathbf{1}_{|S|}, \pi_s^a \geq 0\}$ is a closed set, we also have $\pi^* \in \{\pi : B^\top \pi = \mathbf{1}_{|S|}, \pi_s^a \geq 0\}$. Moreover, since $\nabla\Phi(\pi)$ goes to infinity as any entry $\pi_s^a \to 0$, we have $(\pi^*)_s^a > 0$ for any $s$ and $a$. Hence $\pi^{(k)}$ is contained in a closed set $K$ contained in $\{\pi : B^\top \pi = \mathbf{1}_{|S|}, \pi_s^a > 0\}$ for large enough $k$. By the assumption that $\nabla f$ is Lipschitz continuous, we have

$$\lim_{k \to \infty} \frac{f(\pi^{(k+1)}) - f(\pi^{(k)}) - \nabla f(\pi^*) \left( \pi^{(k+1)} - \pi^{(k)} \right)}{\|\pi^{(k+1)} - \pi^{(k)}\|} = 0. \tag{40}$$

Moreover, since $f(\pi)_{sa} = -(r_s^a - ((I - \gamma P^a)v_\pi)_s)$, which has a similar form with $E(\pi)$, we can directly obtain $\nabla f(\pi)$:

$$(\nabla f(\pi))_{sa,tb} = \lambda_{sa,t}(\pi)\left( -f(\pi)_{tb} + \tilde{c}(\pi)_t - \nabla\Phi(\pi)_{tb} \right) \tag{41}$$

where $\lambda_{sa,t}(\pi) = Z_\pi^{-\top}\tilde{e}_{sa}$ and $\tilde{e}_{sa}$ is the $s$-th row of $I - \gamma P^a$. Notice that at $\pi^*$ we have $\frac{d(\nabla\Phi(\pi))}{dt} = 0$, so $f(\pi^*) + \nabla\Phi(\pi^*) = B^\top c(\pi^*)$, thus

$$
\begin{aligned}
&(\nabla f(\pi^*)(\pi^{(k+1)} - \pi^{(k)}))_{sa} \\
&= \sum_{tb} \lambda_{sa,t}(\pi^*)\left(-f(\pi^*)_{tb} + \tilde{c}(\pi^*)_t - \nabla\Phi(\pi^*)_{tb}\right)(\pi_{tb}^{(k+1)} - \pi_{tb}^{(k)}) \\
&= \sum_{tb} \lambda_{sa,t}(\pi^*)\left(\tilde{c}(\pi^*)_t - c(\pi^*)_t\right)(\pi_{tb}^{(k+1)} - \pi_{tb}^{(k)}) \\
&= \sum_{t}\left[\left(\lambda_{sa,t}(\pi^*)\left(\tilde{c}(\pi^*)_t - c(\pi^*)_t\right)\right)\left(\sum_{b}(\pi_{tb}^{(k+1)} - \pi_{tb}^{(k)})\right)\right] \\
&= 0,
\end{aligned}
\tag{42}
$$

where the last equality results from the fact that $\sum_b \pi_{tb}^{(k+1)} = \sum_b \pi_{tb}^{(k)} = 1$ for any $t$. With (40) and (42), we arrive at

$$
\lim_{k\to\infty} \frac{f(\pi^{(k+1)}) + \nabla\Phi(\pi^{(k+1)}) - B^\top c(\pi^{(k)})}{\|\pi^{(k+1)} - \pi^{(k)}\|} = 0.
\tag{43}
$$

Multiply the fraction above by the unit vector $\frac{\pi^{(k+2)} - \pi^{(k+1)}}{\|\pi^{(k+2)} - \pi^{(k+1)}\|}$, we obtain

$$
\begin{aligned}
&\frac{\left(f(\pi^{(k+1)}) + \nabla\Phi(\pi^{(k+1)}) - B^\top c(\pi^{(k)})\right)^\top (\pi^{(k+2)} - \pi^{(k+1)})}{\|\pi^{(k+1)} - \pi^{(k)}\|\|\pi^{(k+2)} - \pi^{(k+1)}\|} \\
&= \frac{\left(f(\pi^{(k+1)}) + \nabla\Phi(\pi^{(k+1)}) - B^\top c(\pi^{(k+1)})\right)^\top (\pi^{(k+2)} - \pi^{(k+1)})}{\|\pi^{(k+1)} - \pi^{(k)}\|\|\pi^{(k+2)} - \pi^{(k+1)}\|} \\
&= \frac{\left(\nabla\Phi(\pi^{(k+2)}) - \nabla\Phi(\pi^{(k+1)})\right)^\top (\pi^{(k+2)} - \pi^{(k+1)})}{\|\pi^{(k+1)} - \pi^{(k)}\|\|\pi^{(k+2)} - \pi^{(k+1)}\|},
\end{aligned}
\tag{44}
$$

where the second equality uses the constraint $B\pi^{(k+1)} = B\pi^{(k+2)} = \mathbf{1}_{|S|}$. Then from (43) we have

$$
\lim_{k\to\infty} \frac{\left(\nabla\Phi(\pi^{(k+2)}) - \nabla\Phi(\pi^{(k+1)})\right)^\top (\pi^{(k+2)} - \pi^{(k+1)})}{\|\pi^{(k+1)} - \pi^{(k)}\|\|\pi^{(k+2)} - \pi^{(k+1)}\|} = 0.
\tag{45}
$$

By direct calculation of $\nabla^2\Phi$, which is diagonal for all the functions $\Phi$ defined in this theorem, we can see that $\nabla^2\Phi$ is lower bounded on $K$, so $\Phi$ is strongly convex. As a result, there is some constant $\rho > 0$ such that

$$
(\nabla\Phi(\pi) - \nabla\Phi(\tilde{\pi}))^\top (\pi - \tilde{\pi}) \geq \rho\|\pi - \tilde{\pi}\|^2
$$

for any $\pi$ and $\tilde{\pi}$ in $K$. Then

$$
0 = \lim_{k\to\infty} \frac{\|\pi^{(k+2)} - \pi^{(k+1)}\|^2}{\|\pi^{(k+1)} - \pi^{(k)}\|\|\pi^{(k+2)} - \pi^{(k+1)}\|} = \lim_{k\to\infty} \frac{\|\pi^{(k+2)} - \pi^{(k+1)}\|}{\|\pi^{(k+1)} - \pi^{(k)}\|},
\tag{46}
$$

from which we can conclude that $\pi^{(k)}$ converges to $\pi^*$ superlinearly, i.e., (50) holds. In fact, for any $\epsilon$ (assume $\epsilon < 1/2$ without loss of generality), there is some $k(\epsilon)$ such that for any $k > k(\epsilon)$,

$$
\frac{\|\pi^{(k+2)} - \pi^{(k+1)}\|}{\|\pi^{(k+1)} - \pi^{(k)}\|} < \epsilon,
$$

then for any $k > k(\epsilon)$

$$
\begin{aligned}
\|\pi^{(k+1)} - \pi^*\| &\leq \sum_{n=k+1}^{\infty} \|\pi^{(n+1)} - \pi^{(n)}\| \leq \sum_{n=k+1}^{\infty} \epsilon^{n-k}\|\pi^{(k+1)} - \pi^{(k)}\| \\
&\leq \frac{\epsilon}{1-\epsilon}\|\pi^{(k+1)} - \pi^{(k)}\| \leq 2\epsilon\|\pi^{(k+1)} - \pi^{(k)}\|.
\end{aligned}
\tag{47}
$$

Then

$$
\begin{aligned}
\|\pi^{(k)} - \pi^*\| &\geq \|\pi^{(k+1)} - \pi^{(k)}\| - \|\pi^{(k+1)} - \pi^*\| \\
&\geq (\frac{1}{2\epsilon} - 1)\|\pi^{(k+1)} - \pi^*\|.
\end{aligned}
\tag{48}
$$

For any $M > 0$, take $\epsilon = 1/(2M + 2)$, then for any $k > f(\epsilon)$,

$$\|\pi^{(k)} - \pi^*\| \geq (\frac{1}{2\epsilon} - 1)\|\pi^{(k+1)} - \pi^*\| = (M + 1)\|\pi^{(k+1)} - \pi^*\| > M\|\pi^{(k+1)} - \pi^*\|, \quad (49)$$

which shows that $\lim_{k \to \infty} \frac{\|\pi^{(k)} - \pi^*\|}{\|\pi^{(k+1)} - \pi^*\|} = +\infty$ and thus

$$\lim_{k \to \infty} \frac{\|\pi^{(k+1)} - \pi^*\|}{\|\pi^{(k)} - \pi^*\|} = 0. \quad (50)$$

Now, from (39) and (42) we have

$$
\begin{aligned}
&f(\pi^{(k+1)}) + \nabla\Phi(\pi^{(k+1)}) - B^\top c(\pi^{(k)}) \\
&= f(\pi^{(k+1)}) - f(\pi^{(k)}) - \nabla f(\pi^*)\left(\pi^{(k+1)} - \pi^{(k)}\right) \\
&= \left(\int_0^1 \left[\nabla f(\pi^{(k)} + t(\pi^{(k+1)} - \pi^{(k)})) - \nabla f(\pi^*)\right] \mathrm{d}t\right)\left(\pi^{(k+1)} - \pi^{(k)}\right) \\
&\leq \tilde{C}\|\pi^{(k)} - \pi^*\|\|\pi^{(k+1)} - \pi^{(k)}\|,
\end{aligned}
\quad (51)
$$

for some constant $\tilde{C}$, where we used (50) and the Lipschitz contiuity of $\nabla f$ in the last equality. Multiply both sides by $\frac{\pi^{(k+2)} - \pi^{(k+1)}}{\|\pi^{(k+2)} - \pi^{(k+1)}\|}$ and by (44) and the strong convexity of $\Phi$, we have

$$
\begin{aligned}
&\rho\|\pi^{(k+2)} - \pi^{(k+1)}\|^2 \\
&\leq \left(\nabla\Phi(\pi^{(k+2)}) - \nabla\Phi(\pi^{(k+1)})\right)^\top (\pi^{(k+2)} - \pi^{(k+1)}) \\
&= \left(f(\pi^{(k+1)}) + \nabla\Phi(\pi^{(k+1)}) - B^\top c(\pi^{(k)})\right)^\top (\pi^{(k+2)} - \pi^{(k+1)}) \\
&\leq \tilde{C}\|\pi^{(k)} - \pi^*\|\|\pi^{(k+1)} - \pi^{(k)}\|\|\pi^{(k+2)} - \pi^{(k+1)}\|,
\end{aligned}
\quad (52)
$$

which implies that

$$\|\pi^{(k+2)} - \pi^{(k+1)}\| \leq \tilde{C}\|\pi^{(k)} - \pi^*\|\|\pi^{(k+1)} - \pi^{(k)}\|. \quad (53)$$

From (50), we have

$$\lim_{k \to \infty} \frac{\|\pi^{(k)} - \pi^{(k+1)}\|}{\|\pi^{(k)} - \pi^*\|} = \lim_{k \to \infty} \frac{\|\pi^{(k+1)} - \pi^{(k+2)}\|}{\|\pi^{(k+1)} - \pi^*\|} = 1. \quad (54)$$

Combine this with (53) we get

$$\|\pi^{(k+1)} - \pi^*\| \leq C\|\pi^{(k)} - \pi^*\|^2, \quad (55)$$

for some constant $C$, which closes the proof.

$\square$

