# OpenReview forum: "Quasi-Newton policy gradient algorithms"
_ICLR.cc/2022/Conference — ICLR 2022 Submitted_

### Official Review · Reviewer_QCBv · 2021-11-02

**Correctness:** 4
**Technical Novelty And Significance:** 2
**Empirical Novelty And Significance:** Not applicable
**Recommendation:** 5
**Confidence:** 3

**Main Review:**

As said in the summary section, this paper proposes a quasi-Newton method that uses the diagonal of the Hessian matrix as an approximate to accelerate the convergence of the policy gradient algorithm. It shows that under certain conditions along with close to the optimal initial policy, the quasi-Newton policy gradient algorithm converges quadratically.

It's really interesting to see the quadratic convergence for the quasi-Newton policy gradient method. But the conditions under which this convergence holds is very unclear. For example, in Theorem 5, the assumptions are the \nabla f is Lipschitz on a closed subset and \theta^{k} \to \theta^{\star}. How could we verify that these conditions are met for certain problems? I think answering this question could help quantify the localness definition for this paper.
For the first example, the initial policy is uniform. Since based on the theorem, its quadratic convergence holds when the initial policy is very close to the optimal one, is this true for this example as well? Also, for the compared algorithms, do you also use the same initial policy? How about the manually tuned learning rate for GPMD?
Cen et al. (2020) also has quadratic convergence result for close to optimal policy case. How does this paper's result differ from it?

**Summary Of The Paper:**

The paper is concerned with infinite horizon discounted (finite state and finite action space) MDP problem with known reward and transition matrices.
To solve this problem, it proposes a quasi-Newton method for the policy gradient algorithm with entropy regularization.
The main contribution is the it proves that the proposed algorithm has local quadratic convergence property under certain assumptions.
It also uses two examples to show that the proposed algorithm(s) is faster than the current policy gradient based algorithms in the literature.


**Summary Of The Review:**

The quadratic convergence of the proposed quasi-newton algorithm is interesting, but the conditions under which it holds is pretty unclear. Since this is the paper's main contribution, it's a borderline submission for me at this moment. I will wait for the rebuttal to address my concerns.

---

### Official Review · Reviewer_eYzw · 2021-11-02

**Correctness:** 3
**Technical Novelty And Significance:** 2
**Empirical Novelty And Significance:** 2
**Recommendation:** 5
**Confidence:** 4

**Main Review:**

Although, I think this is an interesting paper I have some questions that I would be grateful if answered:

1. I found the exposition of the paper to be a bit confusing and not very clear. Is it possible for the authors to clearly list in a sub-section (even in the appendix) the assumptions used when constructing the proofs?
2. Can the authors kindly illustrate the novelty conveyed in the proof. In other words, how is this analysis novel compared to standard analysis of quasi-newton methods beyond its application to entropy regularised reinforcement learning? That is not to say that the analysis is not rigorous, I just want to understand if there were any hurdles that needed to be overcome carrying optimisation proofs to RL settings.
3. Concerning the assumptions, is this analysis assuming convexity? Of course, in discrete cases, this can be met but it is not general. If so, I think there needs to be a section clearly elaborating the limitations of the current proof.
4. Could the authors report the running time of their algorithm rather than just demonstrating iterations? It would be great if those running times are also compared to standard policy gradients and Natural policy gradients.
5. Could the authors please show reward learning curves rather than just policy changes or log errors?
6. What does an eta=1 really mean? I do understand that setting a learning rate that high can lead us to the statement of the proof but what does it mean emprically?


**Summary Of The Paper:**

In this paper, the authors propose a quasi-Newton method for policy gradient algorithms in reinforcement learning while being entropy regularised. Their method acts as an umbrella of other techniques including Natural policy gradients. Interestingly, upon using different regularisers, the approach yields new policy gradient algorithms.

**Summary Of The Review:**

Please see above.

---

### Official Review · Reviewer_Wec3 · 2021-11-02

**Correctness:** 4
**Technical Novelty And Significance:** 3
**Empirical Novelty And Significance:** 3
**Recommendation:** 5
**Confidence:** 3

**Main Review:**

Overall I enjoyed reading this paper. The presentation is clear, but the writing can be improved, as explained in the minor comments.

Major Comments:

(1) Since using entropy regularization changes the problem, $\pi^*$ of the modified problem is no longer the true optimal policy of the original problem. To complete the story, it is better to provide a bound on the difference between $\pi^*$ of the modified problem and the true optimal policy? As a follow-up question, I believe there is a trade-off between the asymptotic error and the convergence rate in choosing the regularization coefficient. Is it possible to make the regularization coefficient time-varying so that we have asymptotic convergence to the true optimal policy, and also have improved convergence rate?

(2) In the statement of Theorem 5, the authors assume that $\theta^k$ converges to $\theta^*$. Since $\theta^k$ is a function of the iterate $\pi^k$ generated by the algorithm, this assumption seems to be problematic. Is it possible to actually prove the statement instead of assuming it? What is the major difficulty there?

(3) Regarding Eq. (18) of Theorem 5, does it hold for all $k$ or only for $k$ large enough? If it is the latter case, what parameters of the problem does the threshold depend on? Regarding the constant $C$, since it is not a numerical constant, what does it depend on?

(4) Newton's method has been studied extensively in the literature. What is the main technical challenge in extending existing results on using Newton's method to solve optimization problem to the setting of policy gradient algorithm?

(5) The policy gradient method is introduced mostly for solving the reinforcement learning (RL) problem, where the MDP model is unknown. Existing algorithms such as PG and NPG can be easily generalized to the RL setting. For the proposed algorithm in this paper, due to the presence of $P^a$, it is not immediately clear how to actually use it in the RL setting, except to first use the model-based approach to estimate $P^a$. It would be nice to have a paragraph discussing about the possibility of extending the results in this paper to the RL setting.

Other Comments:

(1) About writing: When stating a Theorem (Proposition, Lemma, etc), it is better to introduce all the notation before the result. In that case, the statement of the result is more concise.

------------------After Author Feedback-------------------

Since the author did not provide any feedback, I would like to keep my score and vote for rejection.


**Summary Of The Paper:**

This paper proposes a quasi-Newton method for policy gradient algorithm with entropy regularization, which is popular in solving the reinforcement learning problem. With various entropy functions, this paper establishes quadratic convergence rate for the proposed algorithm. Such convergence rate is verified using numerical experiments.

**Summary Of The Review:**

Overall I think this paper is interesting and has the potential to contribute theoretically to policy gradient method. There is some additional work (see my main review) that needs to be done to make this paper more competitive.

---

### Official Review · Reviewer_RnjS · 2021-11-05

**Correctness:** 3
**Technical Novelty And Significance:** 2
**Empirical Novelty And Significance:** 1
**Recommendation:** 3
**Confidence:** 3

**Main Review:**

The primary strength of this paper is the characterization of the Hessian around the optimal policy for a tabular policy representation. This analysis may be helpful to others who do theoretical analysis on policy gradient methods.

The biggest issue is with this paper is that it does not offer clear support that this Hessian approximation is close when the policy is not trivially close to the optimal policy. There is no investigation as to what effect this precondition matrix has on the optimization surface. Since the resulting update is simply a diagonalized version of the natural policy gradient method, one would expect further insight into when and why the method is advantageous.

These experiments have two issues. The first is that the comparison between methods is unfair. The same step size is used for all methods, but they all have different dynamics, and it is not clear that the same hyperparameters for each method provide a meaningful comparison. The second is that the experiments shed no light on the type of problems this method will solve efficiently. The methods are only tested on two MDPs, and there is no discussion as to what characteristics of the policy optimization problem these problems are designed to test.

The paper also makes it seem like it is addressing policy optimization issues for typical RL settings, but the setting of this paper is not typical in practice. First, it does not acknowledge that these results do not pertain to cases with function approximation, which is the more common case of policy optimization. Second, it assumes full knowledge of the reward function and transition dynamics, which is rarely the case. In this setting, it is not clear that policy gradient methods offer any advantage over value iteration policy iteration techniques.


Questions:
Is the following a correct interpretation of the approximate Hessian? The approximate Hessian assumes that the dominant terms of the Hessian only depend on the diagonal of the Fisher information matrix (FIM) of the policy parameters. This interpretation implies that off-diagonal components of the FIM and any terms that are influenced by the reward function have little impact on the curvature of the objective function.

How close does the policy need to be optimal before this Hessian approximation is reasonable?

Does the approximate Hessian provide a helpful update direction and step length when the policy is not nearly optimal?

Why would one use this Hessian approximation instead of the diagonal of the Gauss-Newton method?

Why are the MDPs in the experiments well suited to demonstrate the essential properties of these policy gradient methods? What are the properties of the MDPs in the experiment that make these policy optimization methods converge quickly?



Corrections:
Section 1.1 first contribution. The algorithm does not recover the natural gradient algorithm but uses the diagonal of the Fisher information matrix.

Section 2: definition of w_\pi -- Z_\pi should have a ^{-1} in the exponent.


Suggestions for improving clarity
Page 1: arbitrary weight vector e. It might be worth mentioning that in most RL problems, this weight vector only covers a small set of possible states, making optimization difficult.

Section 2: r_\epsilon and Z_\epsilon are introduced, but it is unclear what role they have or why they are being introduced.

Section 2: Formulation of the policy gradient is not immediately obvious, and no derivation or reference is given for the expression.

Section 2: Measure for O(||\pi - \pi^*||) is not given until much later in the paper. It was not clear if this was meant to be a norm or divergence measure from the writing.




**Summary Of The Paper:**

This paper develops an approximate second-order policy optimization method to overcome the slow convergence of policy gradient methods. This paper proposes an entropy regularized version policy gradient method that approximates the Newton update using a diagonal preconditioner. Theoretical results a presented that show the convergence rate of the new method near the optimal solution is quadratic. A generalized version of the algorithm is presented for any entropic regularizer. Experiments on two tabular MDPs demonstrate that the algorithm can reach convergence in a single-digit number of iterations.



**Summary Of The Review:**

I do not recommend this paper for acceptance because there are many essential unanswered questions regarding the proposed method.

---

### Decision · Program_Chairs · 2022-01-20

**Decision:**

Reject

**Comment:**

This is a nice paper which shows that KL-regularized natural policy gradient (assuming exact access to the MDP, meaning no noise in the reward and Q function estimates), which achieves linear convergence, can use ideas from quasi-newton methods and recover their quadratic convergence.  Given the excitement surrounding policy gradient methods and their convergence rates, this is a valuable direction and family of ideas.  Unfortunately, the reviewers had many concerns about presentation, and also of the exact meaning and relationship of the results to prior work; I'll add to this and note that one issue with quasi-newton methods is that it is unclear how long the "burn-in" phase is, meaning the phase before their quadratic convergence kicks in, and this is still an issue in the present work's theory; another issue, as raised by reviewers, is the difference between the regularized and unregularized optimal policies.  As such, it makes sense for this paper to receive more time and polish.